# Simulation of Figures of Merit for Barristor Based on Graphene/Insulator Junction

**DOI:** 10.3390/nano12173029

**Published:** 2022-08-31

**Authors:** Jun-Ho Lee, Inchul Choi, Nae Bong Jeong, Minjeong Kim, Jaeho Yu, Sung Ho Jhang, Hyun-Jong Chung

**Affiliations:** Department of Physics, Konkuk University, Seoul 05029, Korea

**Keywords:** graphene, barristor, Fowler–Nordheim tunneling, cut-off frequency, delay time, power-delay product

## Abstract

We investigated the tunneling of graphene/insulator/metal heterojunctions by revising the Tsu–Esaki model of Fowler–Nordheim tunneling and direct tunneling current. Notably, the revised equations for both tunneling currents are proportional to *V*^3^, which originates from the linear dispersion of graphene. We developed a simulation tool by adopting revised tunneling equations using MATLAB. Thereafter, we optimized the device performance of the field-emission barristor by engineering the barrier height and thickness to improve the delay time, cut-off frequency, and power-delay product.

## 1. Introduction

Graphene barristor [GB] has been introduced to break the limitation of the low *I*_ON_/*I*_OFF_ of the graphene field-effect transistor (GFET) [1]. Since the introduction of the barristor, not only Si, but either organic [2,3,4,5] or inorganic [6,7,8,9,10,11,12] materials, have been exploited to make the graphene-semiconductor junction and applied on photosensors, gas sensors, etc., with its superior *I*_ON_/*I*_OFF_.

Extreme temperature changes or degradation due to the environment may affect the performance of GFET or GB [13]. However, graphene-insulator-junction barristors exhibit a more stable performance compared to semiconductor-based ones [14].

In this study, we revised the models for Fowler–Nordheim tunneling (FNT) and direct tunneling (DT) in graphene/insulator/metal (GIM) junctions using the Tsu–Esaki tunneling model to reflect the graphene’s linear band structure [15]. Compared to the traditional FNT equation—proportional to *V*^2^—the revised FNT equation—proportional to *V*^3^—fits better with the experimental data. Then, we simulated how delay time (*τ*), power-delay product (PDP), and cut-off frequency (*f*_T_) of the field-emission barristor (FEB) could be improved by varying the tunneling-barrier height ($\emptyset_{B}$), and the thickness of the hexagonal boron nitride (hBN) (*t*_Tunnel_) with the revised tunneling models. We also considered thermionic emission for the low barrier height and DT for the thin insulator channel. The figures of merit of the FEB were extracted from the experimentally measured *I–V* characteristics in [14]. Notably, we could improve the device performance by decreasing *t*_Tunnel_. This is because $\emptyset_{B}$, followed by the channel current (*I*_D_), decreased with *t*_Tunnel_. However, the improvement in the device performance by increasing *t*_Tunnel_ has a limitation because the increase in *t*_Tunnel_ deteriorates the PDP. Therefore, to not only improve *τ* and *f*_T_ but also PDP, both the tunneling barrier height and *t*_Tunnel_ should be decreased.

## 2. Materials and Methods

We used the polydimethylsiloxane (PDMS) stamping method to create the bottom structure (gate electrode/hBN). Few-layer hBN was prepared on PDMS (PF Film-X4-6.5 mil bought in Gel-Film^®^) by mechanical exfoliation, and it was attached to the slide glass upside down. The hBN was subsequently aligned on top of the target substrate using a 3-axes manipulator. The PDMS was heated to 60 °C while it was transferred onto the target. Finally, PDMS was mechanically peeled off after the temperature was decreased to 25 °C. The top structure (hBN/graphene) was then transferred to the metal/hBN structure using the PMMA transfer method. The drain and source electrodes were deposited using an e-beam evaporator. The processes were conducted at the Core Facility Center for Quantum Characterization/Analysis of Two-Dimensional Materials and Heterostructures.

The simulation data were calculated using MATLAB R2021b. The source code implements the equations derived in the manuscript. We utilized MATLAB to calculate the current by varying the barrier height and the electric field. Using the double for loops, we obtained the current ranging from the height up to 3.0 eV and the field up to 0.4 V/nm with the discrete tunneling thickness. The source code is provided in the Appendix A.

## 3. Results

### 3.1. Tunneling Current Model in GIM Junction

We derived a revised model for tunneling in graphene/insulator/metal (GIM) junctions. We started with the Tsu–Esaki model and applied the band structure of graphene. The current density tunneling from graphene to the metal can be calculated as follows [16]:(1)$$dJ_{g\to m}=qv_{x}T^{\prime }\left(k_{x}\right)D_{g}\left(k_{g}\right)f_{g}\left(E\right)\left(1-f_{m}\left(E\right)\right)dk_{x}dk_{g},$$
where q is the elementary charge; $v_{x}$ is the velocity in the x-direction, which is perpendicular to the graphene plane (to the insulator); $T^{\prime }\left(k_{x}\right)$ is the transmission coefficient; $k_{g}$ and $k_{x}$ are the wave vectors of the carrier parallel and perpendicular to the graphene plane (y-z plane), respectively; and $D_{g}\left(k_{g}\right)$ is the density of states of the graphene with momentum $k_{g}$. In addition, $f_{g}\left(E\right)$ and $1-f_{m}\left(E\right)$ indicate the Fermi–Dirac distribution of the filled states in the graphene and empty states in the metal, respectively. The density of states $D_{g}\left(k_{g}\right)$ can be obtained as $D_{g}\left(k_{g}\right)dk_{g}=D_{g}^{*}\left(k_{y},k_{z}\right)dk_{y}dk_{z}{|}_{k=k_{g}}$, where $k=\sqrt{k_{x}{}^{2}+k_{y}{}^{2}}$ and $D_{g}^{*}\left(k_{y},k_{z}\right)$ is the number of states per unit cell in the 2D momentum space. The areas of the primitive cell (*S*_1_) in real space and reciprocal lattice space (*S*_2_) of graphene are $S_{1}=\frac{\sqrt{3}}{2}a^{2}$ and $S_{2}=\frac{8\pi^{2}}{\sqrt{3}a^{2}}$, respectively (Appendix A) $D_{g}^{*}\left(k_{y},k_{z}\right)=2*2*\frac{1}{S_{1}}*\frac{1}{S_{2}}=\frac{1}{\pi^{2}}$, considering the spin and valley degeneracy [17,18]. Thus, $D_{g}\left(k_{g}\right)dk_{g}=\frac{1}{\pi^{2}}2\pi k_{g}dk_{g}=\frac{2}{\pi }k_{g}dk_{g}$, as shown in Appendix A. Since graphene has a linear dispersion relation around the K-point, so that $E_{g}=\hbar v_{F}k_{g}$ or $v_{F}dk_{g}=\frac{1}{\hbar }dE_{g}$ (inset of Appendix A), and the density of states in energy space $D_{g}\left(E_{g}\right)dE_{g}=\frac{2}{\pi }\frac{E_{g}dE_{g}}{{\left(\hbar v_{F}\right)}^{2}}$, where $v_{F}$ is the Fermi velocity and ℏ is the reduced Plank’s constant.

Using the parabolic dispersion relation so that $E_{x}=\frac{\;\hbar^{2}k_{x}{}^{2}}{2m}$, $\frac{1}{\;\hbar }dE_{x}=v_{x}dk_{x}$, Equation (1) can be re-expressed with $k_{x}$ changed to $E_{x}$ as follows: (2a)$$dJ_{g\to m}=qT^{\prime }\left(E_{x}\right)f_{g}\left(E\right)\left(1-f_{m}\left(E\right)\right)\frac{2}{\pi \hbar^{3}v_{F}{}^{2}}E_{g}dE_{g}dE_{x}$$

As the total energy $E=E_{x}+E_{g}$, Equation (2a) can be re-expressed as:(2b)$$dJ_{g\to m}=qT^{\prime }\left(E_{x}\right)f_{g}\left(E\right)\left(1-f_{m}\left(E\right)\right)\frac{2}{\pi \hbar^{3}v_{F}{}^{2}}\left(E-E_{x}\right)dEdE_{x}$$

Therefore, considering that $E_{g}>0$,
(3)$$J_{g\to m}=\frac{2q}{\pi \hbar^{3}v_{F}{}^{2}}\int_{0}^{\infty }dE_{x}T^{\prime }\left(E_{x}\right)\int_{E_{x}}^{\infty }dEf_{g}\left(E\right)\left(1-f_{m}\left(E\right)\right)\left(E-E_{x}\right)$$

Notably, $\left(E-E_{x}\right)$ of the integrand originates from the linear dispersion relation of graphene, which does not exist in the original transport equation. 

Equation (3) is a model of the tunneling current density in a graphene/insulator/metal junction. It comprises two integral parts: the first integral indicates the tunneling probability between graphene and the metal. Because it includes the tunneling coefficient *T,* which is related to the barrier height and width, it has different forms depending on the tunneling mechanisms such as FNT and DT [19,20,21,22]. The difference will be described in the next subsection. The second integral is related to the carriers supplied at the interface by an applied voltage. Thus, it has an identical form for both the FNT and DT. At the temperature *T* = 0, the integrand is finite only in the interval of $\left[E_{x},E_{F,g}\right]$ because the carrier tunnels only from the filled state of graphene and the empty state of the metal, as shown in Figure 1b,c. Thus, the second integral can be simplified to $\int_{E_{x}}^{E_{F,g}}(E-E_{x})dE=\frac{1}{2}{\left(E_{F,g}-E_{x}\right)}^{2}$, where $E_{F,m}$ is the Fermi level of the metal. Therefore, $J_{g\to m}$ can be written as *T* = 0, as follows:(4)$$J_{g\to m}=\frac{q}{\pi \hbar^{3}v_{F}^{2}}\int_{E_{F,m}}^{E_{F,g}}T^{\prime }\left(E_{x}\right){\left(E_{F,g}-E_{x}\right)}^{2}dE_{x},$$
where is the interval limit between the filled state of graphene and empty state of the metal, and $E_{F,g}$ is the Fermi level of graphene.

While the total current density is the difference between the current tunneling from graphene to metal ($J_{g\to m}$) and that from metal to graphene ($J_{m\to g}$), the total current density J of the GIM junction becomes $J_{g\to m}$ because $J_{m\to g}$ does not contribute to J. This is because $\int_{E_{x}}^{E_{F,g}}f_{m}\left(E\right)\left(1-f_{g}\left(E\right)\right)dE$ becomes zero because $f_{m}\left(E\right)=0$ under $E_{F,m}\leq E$.

### 3.2. Fowler–Nordheim Tunneling Model in GIM Junction

In Equation (4), the transmission coefficient $T\left(E_{x}\right)\;$for FNT in the Wentzel–Kramers–Brillouin approximation can be expressed as $T\left(E_{x}\right)=\exp\left(-\frac{4\pi }{h}\int_{0}^{x_{1}}\sqrt{2m^{*}\left(E_{c}-E_{x}\right)}dx\right)$ [16,19], where *m*∗ is the electron effective mass, the conduction band minimum (CBM) $E_{C}\left(x\right)=E_{F,g}+q\emptyset_{B}-q\frac{V}{d}x$, and $x_{1}=\frac{\left(E_{F,g}+q\emptyset_{B}-E_{x}\right)d}{qV}$, as shown in Figure 1b. Assuming that the work function of graphene and metal are the same, $V=(E_{F,g}-E_{F,m})/q$, where *V* is the potential bias between the graphene and metal. Then, the transmission coefficient can be rewritten as follows:(5)T(Ex)=exp(−4π2m∗h∫0x1dxEF,g+q∅B−qVdx−Ex )  =exp[8π2m∗d3hqV{(EF,g+q∅B−qVdx1−Ex)32−(EF,g+q∅B−Ex)32}].

Since the first term in the curly bracket disappears because of the value of *x*_1_, the tunneling coefficient can be described as follows:(6)T(Ex)=exp(−8π2m∗d3hqV(q∅B−(Ex−EF,g))32)

Then, Equation (4) can be rewritten for the FNT current when *E_x_* is between $E_{F,m}$, and $E_{F,g}$ as follows:(7)J=qπℏ3vF2∫EF,mEF,gexp(−8π2m∗d3hqV(q∅B−(Ex−EF,g))32)(EF,g−Ex)2dEx

The exponential term in Equation (7) can be rewritten by using the following Taylor series as (q∅B−(Ex−EF,g))32 ≅ (q∅B)32−32(Ex−EF,g)(q∅B)12. 

Then, by substituting $E_{x}-E_{F,g}$ with *E,* the FNT current J is given by the following equation:(8a)J=qπℏ3vF2exp(−8π2m∗d3hqV(q∅B)32)∫EF,m−EF,g0exp(E4π2m∗dhqV(q∅B)12)E2 dE

The integral part can be simplified as
(8b)$$\int_{k}^{0}\exp\left(AE\right)E^{2}dE=\frac{1}{A^{3}}\left[2-\exp\left(Ak\right)\left(A^{2}k^{2}\right)-2Ak+2\right)]$$
where A=4π2m∗dhqV(q∅B)12 and $k=E_{F,m}-E_{F,g}$.

Assuming that $E_{F,g}\gg E_{F,m}$, the *k* is very small, as is *Ak*. Therefore, only the constant in Equation (8b) survives. Equation (8a) can then be rewritten as follows:(9)J=q4V34πvF2(2m∗)32(q∅B)32d3exp(−8π2m∗d3hqV(q∅B)32).

Equation (9) represents a new model for the FNT current in a graphene/insulator/metal junction [15]. Notably, in the revised model, the FNT is proportional to *V*^3^, whereas the FNT in the Tsu–Esaki model is proportional to *V*^2^. 

### 3.3. Direct Tunneling Model in GIM Junction

*T*(*E*_x_) for DT can be expressed from Equation (4) as follows [23]:(10)T(Ex)=exp(−4π2m∗h∫0ddxEF,g+q∅B−qVdx−Ex )  =exp[8π2m∗d3hqV{(EF,g+q∅B−qV−Ex)32−(EF,g+q∅B−Ex)32}]

Note that the integration interval for DT is [0, d], as shown in Figure 1c. After integrating and taking the terms up to the order of (q∅B)32 of the Taylor series, the current density for DT can be rewritten as follows:(11)J=qπℏ3vF2∫EF,m−EF,g0exp(4π2m∗dhqVE{(q∅B)12−(q∅B−qV)12})E2 dE,
where *E_x_* is located between $E_{F,m}$ and $E_{F,g}$, $qV\ll q\emptyset_{B}$, and $E_{x}-E_{F,g}$ is replaced by *E*. Therefore, the DT current can be rewritten as follows: (12)J=q4V34πvF2(2m∗)32d3{(q∅B)12−(q∅B−qV)12}3 exp[−8π2m∗d3hqV{(q∅B)32−(q∅B−qV)32}]

Equation (12) is a revised DT model for a graphene/insulator/metal junction. Compared to the DT equation based on the Tsu–Esaki model in the metal/insulator/metal junctions [15], the tunneling current in the Tsu–Esaki model is proportional to *V*^2^, whereas the current in the revised model is proportional to *V*^3^.

### 3.4. FNT Barrier Height

To calculate the barrier height using the FNT equation for the graphene/insulator junction, the FNT equation can be rewritten as follows: (13)$$ln\left(\frac{I_{D}}{V_{D}^{3}}\right)=\gamma +\beta \frac{1}{V_{D}}\;$$
where *I*_D_ is the drain current; *V*_D_ is the drain voltage, γ=lnAeffq44πvF2(2m∗)32(q∅B)32d3 and β=−8π2m∗d3hqV (q∅B)32. In the new equation, *γ* is replaced with $\alpha =ln\frac{A_{eff}q^{3}m}{8\pi h\emptyset_{B}d^{2}m^{*}}$ from the original FNT equation because they are extracted from different second integrations in Equation (3), which are related to the density of states in the metal and graphene, respectively. However, *β* does not change because it is extracted from the tunneling coefficient, which is the first integration in Equation (3). Consequently, the results of the barrier height calculation using the original and revised FNT equations were not significantly different. 

Figure 2a shows the replotted *I*_D_–*V*_D_ curve of the FEB using the revised FNT equation. The graph fits well to the straight line, and its slope was estimated to be −683 V. Figure 2b shows a replotted *I*_D_–*V*_D_ curve obtained using the traditional FNT equation. The slope of the straight line was estimated to be −689 V. The tunneling barrier heights extracted from the slopes were 2.10 eV (by the revised FNT equation) and 2.11 eV (by the traditional FNT equation), respectively. Therefore, the results of the barrier height calculation using the original and revised FNT equation were similar because they used an identical *β* [19,20,21,22,23,24].

However, the discrepancy between the two models became apparent when we estimated the FNT current in the graphene/insulator junctions. In contrast to the barrier calculation, *α* and *γ* affected the tunneling current estimated using each FNT equation. Figure 2c shows the experimental data and simulated *I*_D_–*V*_D_ curves obtained using the original (blue) and revised (red) FNT equations. The black curve indicates the measured *I*_D_–*V*_D_. The red line was estimated using the revised FNT equation: ID,revised=VD3exp[γ−8π2m∗d3hqVD(q∅B)32]. The blue line was obtained by the original FNT equation: ID,original=VD2exp[α−8π2m∗d3hqVD(q∅B)32] [25]. The barrier heights of 2.10 eV and 2.11 eV were applied to the revised and traditional FNT equations, respectively. Although the barrier heights were similar, the simulated currents were significantly different because of the α and γ. As shown in Figure 2c, the red curve simulated by the revised FNT equation was better fitted to the experimental data. Therefore, we used the revised FNT equation to simulate the figures of merit for the FEB.

### 3.5. Simulation for Barrier Height Engineering to Improve Delay Time and Cut-Off Frequency

To evaluate the performance of the FEB, we extracted *τ* and *f*_T_ from the experimental *I–V* curve, where *τ* is a time delay required to charge the gate electrode with *I*_ON_, and *f*_T_ is a maximum frequency up to which the current of the transistor could be amplified [14]. We obtained a delay time of 154 ns and a cut-off frequency of 13.8 MHz for the FEB. Compared to the performance of the graphene/Si barristor, which was simulated using NanoTCAD ViDES (Device simulator) [26], the FEB’s delay time was 140 times slower than that of the graphene/Si barristor (1.1 ns), and the cut-off frequency was 92 times lower than that of the graphene/Si barristor (1.3 GHz). The low performance of the FEB originated from the low ON current (*I*_ON_) because the delay time and cut-off frequency depend on *I*_ON_. In the FEB, the tunneling barrier height should be decreased to improve *I*_ON_. Therefore, to obtain the minimum barrier height in our device, we estimated the drain current by varying the barrier height and electric field strength between the source and drain electrodes (*E*_Field_) using the revised FNT equation.

Figure 3a shows the FNT current simulated by varying $\emptyset_{B}$ and *E*_Field_. To increase the *J*_ON_, a higher *E*_Field_ and a lower $\emptyset_{B}$ are required. However, Figure 3b describes that $\Delta J_{D}∕\Delta \emptyset_{B}$ ratios (slope of lines) decreased with *E*_Field_. This indicates that the device requires more charge on graphene to modulate its work function. Increasing *E*_Field_ to improve *J*_ON_ requires more switching energy for the device. Therefore, the maximum *E*_Field_ should be determined by considering both the on-state current and the energy consumption for switching. Likewise, decreasing the tunneling barrier height is limited by thermionic emission. The thermionic emission current was estimated using the following equation:(14)$$J_{Thermionic}=\frac{qk_{B}^{3}}{\pi \hbar^{3}v_{F}^{2}}T^{3}\exp\left(-\frac{q\emptyset_{B}}{k_{B}T}\right)$$
where *k*_B_ is the Boltzmann constant and *T* is the temperature [27].

Because the thermionic emission current exponentially depends on $\emptyset_{B}$ and the temperature, the total channel current under *E*_Field_ lower than 0.1 V/nm is affected by the thermionic emission current, as shown in Figure 3b. Therefore, an *E*_Field_ above 0.1 V/nm should be applied to avoid temperature dependence of the *I*_D_–*V*_D_ characteristics. As shown in Figure 3b, these conditions for improving the delay time and cut-off frequency were satisfied when the *E*_Field_ was 0.2 V/nm, *J*_ON_ was 10^−4^ A/μm^2^ with $\emptyset_{B}$ 0.5 eV, and *J*_Off_ was 10^−10^ A/μm^2^ with $\emptyset_{B}$ 1.055 eV. The required charge (Q) to decrease $\emptyset_{B}$ from 1.055 eV to 0.505 eV was calculated by using the equation: $∆w_{G}=\frac{h}{2\pi }v_{F}\sqrt{\pi Q}$, where $∆w_{G}$ is the work function shift of graphene [28]. The delay time and cut-off frequency were calculated using the following equations [29]: (15)$$\begin{array}{c} \tau =\frac{Q_{\mathrm{on}}-Q_{\mathrm{off}}}{J_{\mathrm{on}}}=\frac{Q}{J_{\mathrm{on}}}\\ f_{T}=\frac{1}{2\pi }\frac{{dJ}_{D}{/dV}_{G}}{{dQ/dV}_{G}}=\frac{1}{2\pi }\frac{{dJ}_{D}}{dQ} \end{array}$$

We obtained a delay time *τ* of 0.18 ns and cut-off frequency *f*_T_ of 3.98 GHz that were better than 1.1 ns and 1.3 GHz in the graphene/Si barristor. Therefore, when the applied *E*_Field_ was 0.2 V/nm and $\emptyset_{B}$ was changed from 0.5 eV to 1.055 eV or vice versa, FEB could achieve the optimized delay time and cut-off frequency.

### 3.6. Simulation for Insulator Thickness Engineering to Improve Power-Delay Product 

The power delay product (PDP) refers to the energy consumed during device switching. We estimated the PDP of the FEB to be 355 fJ/μm^2^ from the *I*_D_–*V*_G_ graph, which was 47 times greater than that of the graphene/Si barristor (7.5 fJ/μm^2^). This high energy consumption for device switching is because the semiconductor-less device requires a high *V*_D_ to control the FNT current. Therefore, to reduce the PDP, *t*_Tunnel_, where the FNT takes place, should be reduced. 

Figure 3d exhibits the DT current by varying *t*_Tunnel_. When *t*_Tunnel_ was thinner than 2 nm, the FNT could not control the channel current because the DT current dominated the FNT current. In contrast, DT was suppressed below the off-state current in the FNT regime when *t*_Tunnel_ was 6 nm. Therefore, when *t*_Tunnel_ was 6 nm, and *E*_Field_ was 0.2 V/nm, we obtained a PDP of 21 fJ/μm^2^ from the following equation:*PDP* = *V*_D_ (*Q*_on_ − *Q*_off_).(16)

Although still greater than that of the graphene/Si barristor (7.4 fJ/μm^2^), the PDP decreased to 6% of the measurement by engineering *t*_Tunnel_.

## 4. Conclusions

In conclusion, we introduce a new model for the FNT and DT currents of graphene/insulator/metal (GIM) heterojunctions. We obtained new models by revising the supply function of the Tsu–Esaki model. Notably, the tunneling current in the revised FNT equation was proportional to *V*^3^. We then extracted the tunneling barrier height in the graphene/insulator junction from the slope of the line in the *I*_D_–*V*_D_ curve replotted with the axes of *ln*(*I*/*V*^3^) and 1/*V*. The barrier height obtained using the revised model was not significantly different from that of the original model. However, the *I*_D_–*V*_D_ curve estimated by the revised FNT fit better with the experimental data than the *I*_D_–*V*_D_ curve simulated by the original FNT. Then, we simulated the *τ*, *f*_T_, and PDP of the FEB by varying $\emptyset_{B}$ and *t*_Tunnel_ by using the revised FNT equation. These significantly improved by decreasing $\emptyset_{B}$ and increasing *E*_Field_. By considering the thermionic emission at low barrier height and energy consumption at the high electric field, we obtained a *τ* of 0.18 ns and *f*_T_ of 3.98 GHz when *E*_Field_ was 0.2 V/nm, and $\emptyset_{B}$ was changed from 0.5 to 1.055 eV or vice versa. We improved the PDP by decreasing the *t*_Tunnel_. As DT exponentially increased as the thickness decreased, we obtained a lower boundary *t*_Tunnel_ of 6 nm, and then the PDP decreased to 17 times lower than the experimental data.

## Figures and Tables

**Figure 1 nanomaterials-12-03029-f001:**
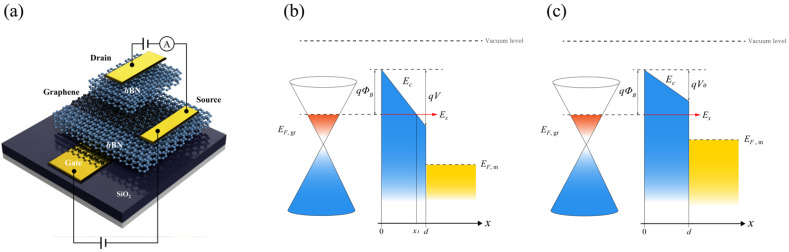
(**a**) A schematic of the field-emission barristor and schematic illustration of the graphene/insulator/metal (GIM) junction in Fowler–Nordheim tunneling; (**b**) an illustration of the FN tunneling. Electron tunnels through the insulator with the tunneling thickness of $x_{1}$; (**c**) an illustration of DT; electrons tunnel through the insulator with a width of *d*.

**Figure 2 nanomaterials-12-03029-f002:**
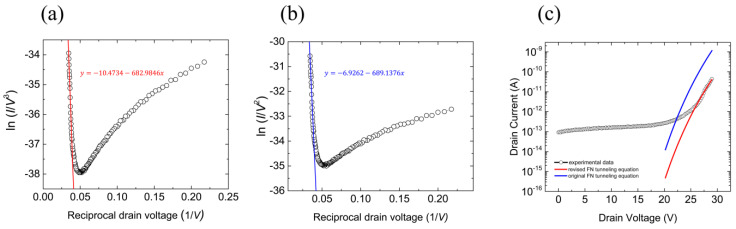
Fitting of FNT to calculate the tunneling barrier height with the revised and traditional equations. (**a**) Straight line (red) fitted to FNT current by the revised FNT equation. Its slope is estimated to be −683 V. A barrier height of 2.10 eV was extracted from the slope. (**b**) Straight line (blue) fitted to the FNT current by the traditional FNT equation. Its slope was estimated to be −689 V. A barrier height of 2.11 eV was extracted from the slope. (**c**) The experimental *I*_D_–*V*_D_ curve (black) of FEB consisting of graphene and hBN, the simulated *I–V* curve by the revised FNT equation (red), and simulated *I*_D_–*V*_D_ curve by the original FNT equation (blue). Further information of fitting method is explained in detail on Appendix B.

**Figure 3 nanomaterials-12-03029-f003:**
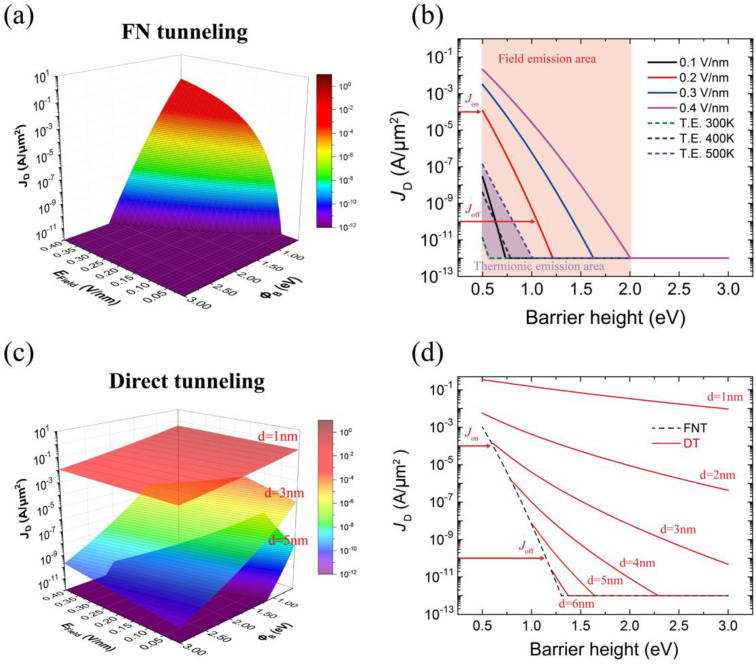
The simulated FNT current as a function of the tunneling barrier height and drain electric field. (**a**) A 3D plot of the FNT equation. (**b**) The channel current under a low drain electric field (0.1 V/nm) can be affected by the thermionic emission current. The Δ*J*_D_/Δ$\emptyset_{B}$ ratios decreased with increasing drain electric field. The on-state current *J*_on_ was 10^−4^ A/μm^2^ when $\emptyset_{B}$ was 0.5 eV, and the off-state current *J*_off_ was 10^−10^ A/μm^2^ when $\emptyset_{B}$ was 1.055 eV. (**c**) A 3D plot of the DT equation for different thicknesses of the insulator. The DT current exponentially increased with a decrease in the thickness. (**d**) The simulated FNT current and DT as a function of the tunneling barrier height for different thicknesses of the insulator. The tunneling current under *E*_Field_ = 0.2 V/nm was simulated by using the revised FNT and revised DT equations. The DT current increased with the decreasing thickness of the insulator. *J*_on_ and *J*_off_ indicate the on-state current and off-state current, respectively.

## Data Availability

Data can be available upon request from the authors.

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
