# Peer review of "Simulation of Figures of Merit for Barristor Based on Graphene/Insulator Junction"

_nanomaterials, 2022, doi:10.3390/nano12173029_

Round 1
Reviewer 1 Report
The Authors presented a model of modified Fowler-Nordheim tunnelling for graphene/insulator/metal heterojunction. Even though the submitted manuscript has no serious scientific issue, there is a problem with the novelty level. The Fowler-Nordheim-like tunnelling in graphene/hBN/metal heterojunction was already reported and described (e.g. J. Appl. Phys. 125, 084902 (2019) or : Nanoscale, 6, 3410 (2014)). The submitted manuscript proposes a modification of standard models; however, the contribution to the present knowledge is not obvious. In other words, if the transition from FW tunnelling into the direct tunnelling is well known and even reported by Authors (Jun-Ho LEE, Hyun-Cheol KIM, Han-Byeol LEE, Nae Bong JEONG, Do-Hyun PARK, Doo-Hua CHOI, Hyun-Jong CHUNG.(2016).Tunneling barristor based on transition of tunneling mode between direct and fowler-nordheim tunneling transport through vertically-stacked Graphene/hBN/Metal.한국진공학회 학술발표회초록집,(),776-776.). As a result, I need to highlight the novelty that proves that it fills gap-in-the-knowledge.
Author Response
We appreciate reviewer 1’s comments on the novelty of the manuscript. The previous studies have studied the Fowler-Nordheim tunneling current from graphene using the traditional current model, which adopted massive electrons in metals.
However, our study focuses on revising the Fowler-Nordheim tunneling current model. We modified the tunneling current from the Tsu-Esaki model to apply the graphene’s linear band structure. Compared to the traditional Fowler-Nordheim tunneling equation, where the current is proportional to V2, the revised model is proportional to V3. We also derived the tunneling current model. Then we optimized the performance of the tunneling devices using the revised current models.
To clarify the point, we revised the introduction in lines 30 – 34, as follows:
In this study, we revised the models for Fowler-Nordheim tunneling (FNT) and direct tunneling (DT) in graphene/insulator/metal (GIM) junctions using the Tsu-Esaki tunneling model to reflect the graphene’s linear band structure [15]. Compared to the traditional FNT equation—proportional to V2, the revised FNT equation—proportional to V3—fits better with the experimental data. Then, we simulated how delay time (τ), power-delay product (PDP), and cut-off frequency (fT) of the field-emission barristor (FEB) can be improved by varying the tunneling-barrier height (φB), and the thickness of the hexagonal boron nitride (hBN) (tTunnel) with the revised tunneling models.
Reviewer 2 Report
In this mansucript authors tackle the tunneling of graphene/insulator/metal
heterojunctions, both experimentally, and by revised theoretical models.
Additionally, using mentioned revised model, they attempted to optimize
barrier height and width (on field emission barristor device) in order to improve device properties making it possibly more suited to real-world applications.
I think that this manuscript is interesting, and contains valuable insights and
results. However, it is not well written, and I believe it doesn't satisfy
journal requirements for publication in the form I received.
In particular:
(1) The English seems to be less than correct on numerous places. Some examples:
- lines 22-25, sentence starting with 'Because': first word should most probably be 'Since';
not to mention that the rest of the sentence is not fully apprehensive
- lines 99-102 (around Eq (4))
- lines 111-113, sentence starting with the 'If'
- lines 115-116
- lines 101-193
... and so on and on.
I suggest that authors consult someone with better English language knowledge to help them write the manuscript.
(2) Line 29: the abbreviation FEB has never be defined; please define it.
(3) Order of the paragraphs in '1. Introduction' is not logical; for example,
it seems that paragraph starting at line 37 should precede on starting on line
29.
(4) The paragraph "2. Material and methods' should be ideal place to put a bit
more details about where and how exactly are MATLAB simulations performed. (It is not clear from manuscript what authors did; or should we study MATLAB code and try to understand what is going on?)
(5) Personally, I would reorganize paragraphs 3.1-3.3 and most of the details
put in appendix, but they are OK here, too.
(6) Figure 2, and linear fits on (a) and (b): this is not how it is done! You
cannot plot data from 0 to 0.25, and then plot fitted lines in range ~0.03-0.04, only. Please, reformat entire figure panel, to show both full data, and separately (as separate figure, or as inset) fitted range, in order to see how well data follows linear behaviour, predicted by theory. The agreement doesn't have to be prefect, but it has to be presented to reader.
(7) Linear fits: calculated slope should have unit, volts (V), shouldn't it?
(8) Lines 191-193
I do not have access to author's previous study [14]; I suppose I could ask
somebody. I appreciate that it is not simple study, however, for the benefit of the reader, authors should include a brief explanation (sentence or two) with important information about the method, or at least clarify which data from present manuscript are used in calculation of delay time and cut-off frequency.
In conclusion, I think there is a lot of potential in this work, but after major
rewrite.
Round 2
Reviewer 1 Report
The Authors modified the manuscript in accordance with comments and/or suggestions, and I have no other option but to support it for publication.